# Alcohol Prevention in Urgent and Emergency Care (APUEC): Development and Evaluation of Workforce Digital Training on Screening, Brief Intervention, and Referral for Treatment

**DOI:** 10.3390/ijerph20227028

**Published:** 2023-11-07

**Authors:** Holly Blake, Emma J. Adams, Wendy J. Chaplin, Lucy Morris, Ikra Mahmood, Michael G. Taylor, Gillian Langmack, Lydia Jones, Philip Miller, Frank Coffey

**Affiliations:** 1School of Health Sciences, University of Nottingham, Nottingham NG7 2HA, UK; emma.adams@nottingham.ac.uk (E.J.A.); wendy.chaplin1@nottingham.ac.uk (W.J.C.); michael.g.taylor@nottingham.ac.uk (M.G.T.); gill.langmack@nottingham.ac.uk (G.L.); lydia.jones@nottingham.ac.uk (L.J.); frank.coffey@nuh.nhs.uk (F.C.); 2NIHR Nottingham Biomedical Research Centre, Nottingham NG7 2UH, UK; 3Department of Research and Education in Emergency Medicine, Nottingham University Hospitals NHS Trust, Nottingham NG7 2UH, UK; lucy.morris@nuh.nhs.uk; 4General Surgery Department, Nottingham University Hospitals NHS Trust, Nottingham NG7 2UH, UK; ikra.mahmood@nuh.nhs.uk; 5Health Innovation East Midlands, Nottingham NG7 2TU, UK; p.miller@nottingham.ac.uk

**Keywords:** health promotion, alcohol, brief intervention, prevention, urgent care, emergency department, digital, health education, workforce, healthcare professionals

## Abstract

Excessive alcohol consumption carries a significant health, social and economic burden. Screening, brief intervention and referral to treatment (SBIRT) is one approach to identifying patients with excessive alcohol consumption and providing interventions to help them reduce their drinking. However, healthcare workers in urgent and emergency care settings do not routinely integrate SBIRT into clinical practice and raise a lack of training as a barrier to SBIRT delivery. Therefore, “Alcohol Prevention in Urgent and Emergency Care” (APUEC) training was developed, delivered, and evaluated. APUEC is a brief, stand-alone, multimedia, interactive digital training package for healthcare workers. The aim of APUEC is to increase positive attitudes, knowledge, confidence and skills related to SBIRT through the provision of (a) education on the impact of alcohol and the role of urgent and emergency care in alcohol prevention, and (b) practical guidance on patient assessment, delivery of brief advice and making referral decisions. Development involved collaborative–participatory design approaches and a rigorous six-step ASPIRE methodology (involving *n* = 28 contributors). APUEC was delivered to healthcare workers who completed an online survey (*n* = 18) and then participated in individual qualitative interviews (*n* = 15). Analysis of data was aligned with Levels 1–3 of the Kirkpatrick Model of Training Evaluation. Survey data showed that all participants (100%) found the training useful and would recommend it to others. Insights from the qualitative data showed that APUEC digital training increases healthcare workers’ perceived knowledge, confidence and skills related to alcohol prevention in urgent and emergency care settings. Participants viewed APUEC to be engaging and relevant to urgent and emergency care workers. This digital training was perceived to be useful for workforce skills development and supporting the implementation of SBIRT in clinical practice. While the impact of APUEC on clinician behaviour and patient outcomes is yet to be tested, APUEC digital training could easily be embedded within education and continuing professional development programmes for healthcare workers and healthcare trainees of any discipline. Ultimately, this may facilitate the integration of SBIRT into routine care and contribute to population health improvement.

## 1. Introduction

### 1.1. Global Burden of Alcohol Consumption

Globally, alcohol use is a leading risk factor for death, injuries and disability [1,2], with significant psychosocial consequences including domestic violence, child abuse, depression and suicide [3]. Data from 195 countries and territories shows that the level of consumption that minimises health loss is zero [4]. The costs associated with alcohol amount to more than 1% of the gross national product in high-income and middle-income countries [5]. The burden of alcohol consumption on healthcare systems in alcohol-consuming countries is estimated to be of a similar or larger order of magnitude than that of the COVID-19 pandemic [3,6]. Despite multiple World Health Organization (WHO) initiatives to reduce alcohol use [7,8], the prevalence of alcohol use has not declined. It is predicted to increase until at least 2030 [9], albeit with geographical variations in the alcohol-attributable burden of disease [10].

There are many effective psychosocial and pharmacological interventions to treat alcohol use disorders (AUDs) and harmful drinking [11]; examples include psychological [12], psychosocial [13], recovery organisations [14], brief interventions [15,16,17], e-interventions [18], mHealth [19], telemedicine [20], mindfulness-based [21], and pharmacological [22]. For people at risk of alcohol-related problems, brief intervention is dominant or cost-effective when compared to no intervention [23]. However, diagnosis and treatment of AUD is often delayed [24]. Globally, only one in six people with AUDs receives treatment [25]. The reasons for delay are complex; a lack of problem awareness [26] and high stigma [24,26,27,28,29,30] can delay help-seeking and service access. There is a need for urgent action to reduce the global burden of alcohol consumption; health promotion is a key aspect of this.

### 1.2. The Need for Alcohol Misuse Prevention in Urgent and Emergency Care Settings

Alcohol consumption contributes to 20% of injury and 11.5% of non-injury emergency presentations [3]. Urgent and emergency care (UEC) settings therefore present a unique environment and “teachable moment” in which to implement health promotion practice, through alcohol screening, brief interventions, and/or referral to treatment (SBIRT) approaches. The aim of brief intervention is to reduce alcohol consumption and related harm in hazardous and harmful drinkers who are not actively seeking help for alcohol problems. Brief intervention is defined as “a conversation comprising five or fewer sessions of brief advice or brief lifestyle counselling and a total duration of less than 60 min” [17]. The conversations usually include feedback on alcohol use, information about the harms and benefits of reducing alcohol intake, and guidance on how to reduce consumption, often focusing on motivational counselling or behaviour change strategies.

There is moderate-quality evidence that brief intervention in emergency settings reduces alcohol consumption in low, moderate [16], hazardous and harmful [17] drinkers, with little additional benefit gained from more extended counselling interventions [17]. It can be a cost-effective approach [31], potentially reducing the negative consequences of alcohol use (e.g., alcohol-related accidents and injuries) [16,32] and the number of repeat visits to emergency departments [16]. However, the integration of SBIRT into routine care is lacking, and there is insufficient systematic screening for alcohol problems in routine healthcare services worldwide [33]. In Australia, among emergency physicians and nurses, only 5% usually formally screen for alcohol problems, 16% conduct brief interventions, and 27% provide a referral to specialist treatment services [34]. In the United States (US), less than one-third of emergency departments offered alcohol brief interventions by trained personnel [35]. There is a need to increase the number of UEC personnel trained in alcohol health promotion practices to support SBIRT delivery in UEC settings.

### 1.3. Barriers to Implementing SBIRT

While healthcare professionals are generally positive towards the concept of health promotion and/or alcohol prevention delivery within UEC settings [36,37,38], and believe it should be routine [38], they raise many barriers to delivery, including lack of knowledge, skills or experience, low motivation, confidence or self-efficacy for implementing SBIRT, perceived lack of time and scepticism of intervention effectiveness [36,39,40]. While single SBIRT contacts during an acute emergency visit have been shown to be acceptable to patients [41], some recipients suggest that the approach, timing, or delivery could be improved [38]. Nonetheless, implementation studies suggest that many of the barriers to the delivery of SBIRT in UEC settings are modifiable [42]. Here, we focus on addressing a specific modifiable factor—the lack of knowledge or skills for SBIRT in UEC workers.

### 1.4. The Need for Training and Education on SBIRT

Training and education of healthcare professionals on alcohol prevention and SBIRT is lacking [35,43] but may help to address many of the commonly raised barriers to implementation. Research has specifically identified a need for SBIRT training amongst the UEC workforce to enhance knowledge, skills, and confidence for SBIRT in UEC settings [36,44,45]. There is currently no training available that is directly targeted to healthcare professionals working in UEC settings and provides guidance on how to deliver SBIRT in these high-pressured and time-sensitive environments. Development of SBIRT training for UEC workers may, therefore, address this gap in healthcare training. As described by Blake and colleagues [46], online training offers many benefits, including low cost (i.e., financial and in-person time), low environmental impact (i.e., reduced travel and printing of materials), consistency and standardisation in delivery, flexibility of use, wide reach, scalability, and greater personal control over learning. Development of a digital training resource on alcohol misuse prevention and SBIRT may, therefore, meet the needs of busy healthcare professionals working in UEC environments.

### 1.5. Study Aim and Research Questions

The overall aim of this study was to develop and test an evidence-based digital workforce training package for UEC workers to facilitate alcohol prevention activities in UEC settings. This digital training is called “Alcohol Prevention in Urgent and Emergency Care” (APUEC) (University of Nottingham, Nottingham, UK). The research questions (RQs) were RQ1: Is APUEC perceived to be relevant and useful to healthcare professionals working in UEC settings?; RQ2: Does APUEC improve users’ attitudes, knowledge, confidence, and skills for SBIRT?; RQ3: Can APUEC contribute to facilitating health promotion practice around alcohol prevention in UEC settings? In this paper, we describe the rigorous methods and approach to the development of the APUEC digital training and report findings of a mixed-methods evaluation that addresses the research questions.

## 2. Materials and Methods

The study adopted a collaborative-participatory design [47] for the development and testing of a digital training package, as used by Blake and colleagues [48,49]. The digital package is a reusable learning object (RLO) developed using ASPIRE methodology [50]. Intervention reporting is guided by the Template for Intervention Description and Replication (TIDieR) Checklist (Appendix A) [51]. The research question was addressed through online survey evaluation mapped to the New World Kirkpatrick Model of training evaluation [52,53]. The study took place during the COVID-19 pandemic, which introduced some delays to development and evaluation due to workload impacts on healthcare workers involved in the study team, peer review panels and evaluation processes. Development activities (*n* = 28) were undertaken between April 2021–March 2022. Delivery of the training (*n* = 18) and survey evaluation (*n* = 18) were completed in April–May 2022. Qualitative interviews (*n* = 15) took place between May and June 2023. This study is part of a wider programme of work on alcohol prevention, for which details are available elsewhere [54].

### 2.1. Reusable Learning Objects

RLOs are “short, self-contained, multimedia web-based resources, including audio, text, images and/or video, and which engage the learner in interactive learning towards a single learning objective or goal” [48]. They take around 15 min to complete and include specific characteristics that enhance learning, including (i) presentation of a concept, fact, process, principle, or procedure; (ii) activities to enhance engagement with content; (iii) self-assessment to apply understanding and test mastery of content; (iv) links and resources to reinforce and support the learning goal [55,56].

### 2.2. ASPIRE Methodology

This is a well-used and validated approach to RLO development [48,50,51,52,53,54,55,56] that is proposed to align directly with the requirements for the design of high-quality training in healthcare [57]. It is based on the principle of participatory co-design and relies on the establishment of a community of practice [58] of experts in the subject area and users from the target audience working in collaboration with instructional designers and multimedia developers. The ASPIRE process consists of six steps: (1) establishing the aims of the RLO (learning outcomes for the target audience), (2) storyboarding (co-creation of content and design), (3) populating/production (translation of ideas into media components), (4) integration (of media components into the platform), (5) release (on a virtual learning environment) and (6) evaluation (of the value of the resource to the target audience). The process is shown in Figure 1, and details for each step are described below. The co-creation approach and engagement of stakeholders throughout the whole development process endeavoured to address RQ1 by ensuring that the materials were relevant and useful (see Step 6 for assessment of RQs1–3).

#### 2.2.1. Step 1: Establishing the Aims

The support need was identified by the project team through discussion with professional networks and reviews of published evidence on alcohol prevention and brief interventions in urgent and emergency care settings [39,44]. The project team had expertise in emergency medicine and nursing, psychology, public health, health promotion, alcohol prevention, brief interventions and behaviour change. Synchronous and asynchronous consultations were held with a virtual expert panel and members of the target audience to establish the key aim and learning outcomes for the RLO. Based on the group discussions and expertise within the project team, the agreed learning objective for this resource was to “increase knowledge, confidence and skills in screening, brief intervention and referral for treatment (SBIRT) for alcohol prevention in an urgent and emergency care settings”. To meet this learning objective, it was agreed that the resource should provide opportunities to learn about (a) the impact of alcohol on individuals and within society, and (b) the role of urgent and emergency care settings in alcohol misuse prevention. This would be achieved through the exploration of how to assess patients’ alcohol consumption, deliver brief advice to patients, and decide when to refer patients for further support or treatment.

#### 2.2.2. Step 2: Storyboarding

A 2 h synchronous online storyboarding event was held remotely using Microsoft Teams (Redmond, Washington, DC, USA), using prepared resources and with real-time facilitator interaction. In total, there were 22 attendees (17 female, 5 male), including members of the project team (*n* = 4), multimedia designers (*n* = 3) and invited individuals with relevant expertise (*n* = 15). The event was led by a health psychologist (H.B.) and an emergency medicine physician (F.C.) and facilitated by two members of the project team (E.A., P.M.) and three multimedia designers from a Health e-Learning and Media (HELM) Team (M.G.T., G.L., L.J.) (School of Health Sciences, Nottingham, UK). The 15 invited attendees (13 female, 2 male) represented four healthcare institutions, bringing expertise in nursing, medicine, public health or emergency services research, and community health protection services (i.e., substance misuse, smoking cessation). Attendees were purposively selected via professional networks to ensure participants represented a range of disciplines relevant to urgent and emergency care, levels of seniority, and settings. This group constituted an expert “community of practice”. The purpose of the event was to co-construct the content, ordering, presentation, and interactive elements that were required for the RLO. At the start of the event, the project team delivered a 45-min introductory presentation to outline (a) the concept of an RLO and development processes (M.G.T., 20 min), (b) the broader subject area of alcohol prevention in UEC (F.C., 10 min), (c) specific RLO topic, objectives and expected output (H.B., 15 min) aligned with three questions (Table 1). Attendees then discussed the questions in small group breakout rooms with an allocated facilitator from the project team and technical support staff from HELM. We used The Mural^®^ (Acme Developer, Inc., San Jose, CA, USA) visual collaboration platform [59], which is a digital interactive whiteboard enabling visual collaboration for teams, to facilitate real-time interaction and recording of discussion outcomes.

#### 2.2.3. Step 3: Populating/Production

Production was undertaken by the project team, which included a public health researcher (E.A.), a health psychologist (H.B.), an emergency medicine physician (F.C.) and three emergency medicine nurses (P.M., L.M., G.M.). Using information gathered in steps 1 and 2, the project researcher populated the RLO content template (specification draft) and worked collaboratively with team members and learning technologists (M.G.T, G.L., L.J.) to review and finalise content, select and develop appropriate graphics and media. We adopted a content template that was recently developed using ASPIRE methodology [48] and replicated the mapping of design principles to RLO design features by Blake and colleagues (Appendix A). The specification was reviewed four times by the project team (July, August, October, and November 2021) and once by learning technologists in the HELM team (October 2021). Content was revised after each review based on feedback from the teams and a final version of the specification was agreed in November 2021. The resulting RLO design allowed users to download a certificate of completion and adapt the media used (e.g., switching text and audio on or off, pausing video, altering the speed of narration) according to personal preferences, contexts, and devices. The final RLO content is shown in Figure 2. Stage 1 peer-review of content (Appendix A) was undertaken with a panel of 10 reviewers, of whom four had attended the initial storyboarding event.

#### 2.2.4. Step 4: Integration

The integration of media components into the platform was undertaken by a learning technologist working collaboratively with the project team. Adopting a mobile-first, design philosophy, the media components of the RLO were integrated using a scalable HTML5 template that maximised user experience across all major platforms and devices. Stage 2 peer review of media and technical presentation (Appendix A) was then undertaken with the same 10 reviewers, with an iterative review of the resource being undertaken by all project team members throughout the process. The final version of the resource was further tested for understandability and functionality with five members of the public. Figure 3 shows screen examples from the final developed RLO. The key revisions and overall findings from the peer review process are shown in Figure 4. Peer reviewers provided a range of minor revisions that were addressed by the project team, examples include: “add clear intended learning outcomes”, “you could refer to the ‘Making Every Contact Count’ approach and use this as a reference source”. They also provided positive feedback: “I thought the tool was well conceptualised, I really love the flow”. The final version included audio narration, and users were able to download a certificate of completion.

#### 2.2.5. Step 5: Release

The final RLO was uploaded to the HELM Open repository, released in January 2022, and made available to users by circulating through professional networks and social media. The URL is available in Appendix A.

#### 2.2.6. Step 6: Evaluation

The evaluation method and analysis adapted the approach reported by Blake and colleagues [48]. Quantitative data were collected in May 2022 via an 18-item survey embedded into the APUEC training package. Survey items (Appendix A) were compiled by the project team and included 10 closed and open-ended items. Item 1 (parts 1–12) was developed by the project team and was specifically related to SBIRT; items 2–10 were adapted from the “Evaluation Toolkit for Reusable Learning Objects and Deployment of e-Learning Resources” [60]. The survey items were aligned with RQ1 (relevance/usefulness). Subsequently, APUEC training was delivered to a convenience sample of 18 healthcare professionals from a single hospital trust in May 2023 as part of a training day for “health improvement champions” at a large teaching hospital trust in England. This group was invited to participate in the evaluation since they held roles that involved health promotion in an acute hospital environment as a core element. All attendees completed APUEC training during the event and were subsequently invited to attend an optional individual qualitative interview specifically focused on gathering their views towards APUEC. The interview topic guide was aligned with the Kirkpatrick model (Appendix A), and items addressed RQs1–3 (relevance/usefulness, attitudes/knowledge/confidence/skills, perceived contribution to health promotion practice). All interviews took place between May and June 2023, online via Microsoft Teams and during working hours. Of 18 training recipients, 15 took part in the interview. Interviews lasted between nine and 21 min (14 min on average) and were conducted by one of four researchers (H.B., W.C., E.A., I.M.). Online informed consent was taken from all interview participants. Participant characteristics (gender, occupation) are shown in Table 2.

Guided by the principles of framework analysis [60], data were mapped to specific indicators on the New World Kirkpatrick Evaluation Model [52] as a theoretical framework, which is a commonly used approach to evaluating the results of training and educational programmes (Figure 5, Table 3). Due to the short timescale between training delivery and interviews, data were collected for Kirkpatrick Levels 1–3 only. Level 4 assessment of impact was not measured in this study since it requires a study with a longer follow-up time to allow for an exploration of how knowledge and skills are implemented in practice and whether they lead to health outcomes.

## 3. Results

APUEC training includes the rationale for alcohol prevention, how to identify and screen patients for alcohol use, how to deliver brief interventions, including communication techniques and behaviour change strategies, and referral for treatment. Overall, this study demonstrated that healthcare professionals were highly satisfied with the training, found it easy to use, and rarely experienced any technical challenges. Participants found the materials engaging and enjoyed the interactive elements, multimedia use, and accessibility of the APUEC. All perceived APUEC as relevant to themselves and others and saw the value of workforce training in influencing health promotion practice and benefiting service users. All participants would recommend APUEC to others. Positive attitudes towards health promotion and SBIRT were reinforced. APUEC improved perceived knowledge, skills, and confidence for SBIRT, particularly for those with less experience in health promotion in UEC environments. Behavioural intentions to practice SBIRT in the future were commonly reported. Findings are reported in detail for each level below.

Based on survey items, post-exposure perceptions of attitudes, knowledge, skills, and confidence to engage in SBIRT are shown in Table 4. Mixed-methods analysis mapping quantitative and qualitative data to the New World Kirkpatrick Evaluation Framework is presented in Table 5, which contains descriptive statistics (from the survey) for Level 1 reach, use and satisfaction, together with illustrative quotes (from the interviews) for every level.

### 3.1. Level 1

The interview participants were highly satisfied with the training, enjoyed using it, and spoke positively about the brief but structured approach of APUEC (“…I’ve done it and it’s fabulous” [ID112, Female, Nurse]). Participants liked the accessibility of the package, including its ease of use, interactivity, and the mixed mediums for the delivery of information (e.g., written text, images, audio narration, video, podcast, and transcripts). They felt the material was engaging and highly relevant to their role in UEC. Only two participants experienced technical issues related to accessing sound on their own device, or challenges with playing the video clip when accessing training on their mobile phone.

### 3.2. Level 2

All interview participants already had high health literacy as a practising healthcare professionals. While this meant that most did not report a change in their attitude after the training (they were already positive towards health promotion), they spoke of the importance of understanding lifestyle choices and how best to support patients who may want to change their behaviour:

*“…we need to start introducing this cultural change in the clinicians’ minds that we don’t just medicate patients for the different symptoms that they come, but we look a little bit deeper into root causes” *.
*[ID105, Male, Doctor]*


Views towards the SBIRT approach to health improvement were positive, with participants advocating for the development of more resources targeting different health areas, such as weight and obesity, smoking, and substance misuse: 

*“I think it just shows that you can make quite a punchy small effect from something small, so there must be able to make other ones, for other situations like drugs, smoking” *.
*[ID107, Female, ACP]*


Interview participants frequently mentioned the value of learning about alcohol screening tools and their ease of use. They reported that the content relating to the number of units of alcohol was useful (“a lot of people, they just don’t know what the cut-offs are” [ID105, Male, Doctor]); this was new learning for some and served as a reminder for others:

*“it helps you ask the right questions to the patients and actually understand the answers that they giving you, because at the moment I think a lot of clinicians, they will say how much alcohol do you drink? They tell them I don’t know, one bottle of wine every other day, but as a clinician you don’t know what that translates to” *. 
*[ID105, Male, Doctor]*


Some participants reported that APUEC had led them to reflect on how much alcohol they consumed themselves, or was consumed by their friends or colleagues (“also for my staff as well, because it’s not just about patients” [ID103, Female, CSW manager]). Many spoke of the value of APUEC in guiding them how to engage in brief interventions that were patient-centred, and flagging where sensitivity was required in opening conversations with patients or clients. They appreciated seeing videos that modelled and gave a structure to these conversations. This provided them with the confidence to have and to practice these conversations with patients:

*“I think if you make it awkward when you’re questioning, the patient’s gonna feel awkward as well. So, it’s just about, think, being confident in your questioning and it’s just saying like I’m gonna be asking you some difficult questions, but I’ve got to kind of ask you about it so you know, sometimes it’s the elephant in the room, isn’t it?” *.
*[ID114, Female, Nurse]*


Most participants expressed their intention to actively promote APUEC training (and therefore engagement with SBIRT) to their colleagues.

### 3.3. Level 3

Three participants reported that they were already employing SBIRT and referring patients for whom they had concerns to an “Alcohol Care Team”. Others reported that they were aware of this referral process. Since the interviews took place soon after exposure to the training, it was not possible in this study to explore the impact of APUEC on changing health promotion practices, per se (“behaviour changes”). However, participants revealed “behavioural intentions” to practice SBIRT in the future. In terms of required drivers, participants commented on who should use SBIRT, approaches for transfer of learning into practice, and when it should happen. Overall, there was a prevailing view that SBIRT could be undertaken by any member of staff with patient contact (i.e., any occupational group), breaking down the barriers of job title (i.e., SBIRT not just to be delivered by those who have health promotion as a key part of their job description), and in any suitable “teachable moment” (i.e., taking advantage of moments in which staff members have already built rapport with a patient).

Participants recognised that the effective transfer of APUEC learning into practice involved an act of “planting a seed”; that is, knowing that the impact may not be immediate but the engagement with SBIRT could potentially make a long-term difference: 

*“it starts the conversation and people have it in the back of their mind…it might take if, like us a few more times, them coming maybe to start the process, ‘cause people might be a bit reluctant or want to start but don’t know, just like there’s some obstacles in the way it might take a while” *. 
*[ID111, Female, Nurse]*


*“So that this becomes more meaningful and impactful in a way that even if the patient says no to me right now, there’s something they will probably go back home and think about it and maybe if they see another healthcare professional, and this topic is again discussed, something springs or kind of you know, just comes up from there and it has a longitudinal impact and positive effect on our patients” *.
*[ID104, Female, Doctor]*


Teamwork was perceived to be a key facilitator of effective SBIRT delivery, which was seen to be an important factor in the contribution of UEC to patient behaviour change and, ultimately, public health (“So maybe they (the patients) can reflect and then seek help if that is what they want” [ID104, Female, Doctor]). 

Participants suggested many routes to implementing APUEC training, including wide dissemination of the web link through email circulation lists, provision of the training at inductions, study days, mentor groups, team-building days, and by reaching out to agency nurses. The broader applicability of training on alcohol prevention was recognised:

*“It’s something that beyond the healthcare sector can actually go into schools, teachers can use them, safety providers can use them. And so it, it transcends beyond the healthcare system itself” *.
*[ID115, Male, Doctor]*


There were divergent views on whether the training should be optional, or mandatory (“it could become what we call mandatory training … then it gets across all staff groups” [ID108, Female, Nurse]; “I wouldn’t really want it to be distilled within, like, you know, mandatory training and become a bit of a cross for people to bear…” [ID102, Male, ACP]. However, having protected time to complete it was commonly raised.

While most participants enjoyed the short, succinct nature of the training as a digital resource, one proposed that the presence of a service user during delivery might be particularly impactful:

*“…someone who’s had lived experience of being helped by an intervention or being helped by a referral process, being helped by a bit of education to add some real potency” *. 
*[ID102, Male, ACP]*


Importantly, interviewees described the importance of the shifting the culture in healthcare to a focus on prevention, rather than treatment alone:

*“I think moving forward we will see more educated patients where they present to their health service, health care services and they want to be consulted on their lifestyle as well and it’s very interesting point where we are because we’re moving from sick care to health care… What can we do to not get sick in the first place?” *.
*[ID105, Male, Doctor]*


This shift requires healthcare organisations to address barriers to implementing SBIRT in UEC environments. Some of the participants, while valuing the APUEC training, highlighted barriers to the implementation of SBIRT in UEC settings. These are primarily related to a lack of time for health promotion, the potential for negative responses from patients, and a lack of privacy in busy clinical environments for raising sensitive issues with patients.

## 4. Discussion

To our knowledge, this is the first study to develop and test an evidence-based digital workforce training package for UEC workers aimed at facilitating alcohol prevention in UEC settings. Our digital training, “Alcohol Prevention in Urgent and Emergency Care” (APUEC), is perceived to be engaging, relevant and useful to healthcare professionals working in UEC settings and improves perceived knowledge, confidence and skills for SBIRT. Workforce training using APUEC is viewed by healthcare professionals to be valuable in facilitating health promotion practice around alcohol misuse prevention in UEC settings. Our study directly responds to prior research identifying a lack of training (and therefore low knowledge, skills, or confidence to engage in SBIRT) as a key barrier to health promotion in UEC settings [36,44].

With regards to engagement with the training, participants in our study valued the usability and accessibility of APUEC. APUEC takes the form of an RLO and is hosted on HELM Open, which is an open-access repository of brief learning resources. All current RLOs on this platform are compliant with the UK Web Content Accessibility Guidelines (WCAG) 2.1 [61], which cover a wide range of recommendations for making Web content more accessible. Including accessibility features is essential for inclusivity; it allows users to customise their learning experience and ensures that all potential users, with and without disabilities, can access the same educational content, engage with the resource, and learn from it. Therefore, in the development of APUEC we considered how to make content accessible on different devices (e.g., desktops, laptops, tablets and mobile devices). APUEC is also designed to be more accessible to people with disabilities (including, but not limited to, accommodations for blindness and low vision, deafness and hearing loss, limited movement, speech disabilities, photosensitivity, and some accommodation for learning disabilities and cognitive limitations). Accessibility features such as transcripts and subtitles are standard in RLO development. Participants in our sample highlighted the benefits of these accessibility features within APUEC digital training for uptake and engagement with the training. The need for accessibility in digital resources is widely acknowledged, and our APUEC training development aligns with other advocates of accessibility, who describe the importance of considering usability, pedagogic issues, varying approaches to learning, technical and resource issues in e-learning development [62]. There is scope to reach a broader audience through the translation of digital training content into other languages. While there are many benefits to online learning, and our participants valued the digital approach, online-only training may not fully address all training needs or preferences, and therefore, a variety of training approaches might be considered, such as online-only, blended learning or face-to-face delivery.

APUEC provides valuable stand-alone digital training on alcohol misuse prevention and SBIRT. With digital training programmes, there is a need to consider potential barriers to technology access and acceptance in the target end-users. In our sample, technical barriers to access were rare and were resolved, with all participants accessing and completing the training. This was facilitated by the simplicity of the route to access (i.e., via web link) and the ability to engage with the training on any device. With regards to barriers to technology acceptance, prior research suggests that perceived usefulness is the most noteworthy factor impacting technology acceptance [63] and 100% of our participants perceived the APUEC training to be helpful, relevant, and would recommend it to others. Therefore, we believe that barriers to technology access and acceptance for this brief training resource are likely to be minimal. However, to maximise uptake of the training in the medium- to long-term, healthcare organisations need to develop plans for training implementation. Reusable learning resources are highly scalable, and our participants suggested numerous routes to sharing the training (i.e., email circulation lists, staff and student inductions, study days, mentor groups, team-building days, via agencies, and continuing professional development programmes). They also proposed that APUEC training could be embedded within the curriculum for healthcare trainees across disciplines. This might require liaison with health education institutions and adoption by professional organisations and bodies; the feasibility and practicality of this requires further investigation. Beyond the ‘uptake’ of the training, it is important to consider how organisations might ‘sustain’ awareness of SBIRT in UEC settings (i.e., the training content) moving forwards. Ongoing activity is likely to be needed to encourage learners to implement SBIRT into their practice. In the first instance, maintenance of awareness might be achieved via regular staff reminders (e.g., emails, handovers, inductions, meetings), active promotion (of APUEC training and SBIRT) to colleagues by dedicated health improvement champions, poster campaigns, or the use of departmental incentives for engagement with health promotion. Future research might consider whether and how different implementation strategies can be used to maximise uptake of digital learning resources. The development of further digital training for UEC workforce may help to raise the profile of health promotion in UEC settings, maintain momentum for prevention activities, and broaden knowledge and skills across diverse occupational groups. Potential topics, as proposed by our participants, might include the wider determinants of health, social prescribing, mental health, smoking cessation, obesity and weight management, physical activity, and drug misuse. Future studies might seek to co-create resources in a range of health areas to generate a repository of RLOs targeting common areas of need in UEC settings. Research could explore the perceived value and relevance to UEC workers and any impacts on healthcare workers’ knowledge, skills, and confidence in health promotion practice in UEC settings.

A cultural shift in healthcare towards prevention is imperative in the context of the increasing prevalence of alcohol use [9], rising pressures on healthcare services due to alcohol use [3,6], and the dramatic, negative impacts of alcohol as a leading risk factor for mortality, morbidity, and adverse psychosocial outcomes [1,2,3]. Research suggests that integrating health promotion, and specifically SBIRT, into UEC environments is viewed positively by many UEC workers [36,37,38] and is acceptable to patients [41]. However, several barriers to SBIRT implementation need to be addressed before healthcare professionals can capitalise on APUEC learning, and the “teachable moments” that consistently arise in UEC settings. Barriers to SBIRT delivery in UEC primarily relate to lack of time (i.e., due to heavy workloads and high service demand), suitability of the physical environment (i.e., over-crowding and lack of privacy in UEC settings), challenges with onward referral systems. Although it was beyond the scope of this research to study the barriers and enablers of SBIRT delivery in any depth, other studies provide insights into the challenges of SBIRT delivery and strategies that are helpful in the implementation process [36,45]. These fundamental structural and job-related barriers to the delivery of prevention in UEC need to be addressed before health promotion will be universally accepted and practiced in UEC settings. In the meantime, APUEC training is a step-change in the provision of workforce training in SBIRT for alcohol misuse prevention for those working in high-pressured and time-sensitive environments. APUEC could be used as a stand-alone training resource or embedded within it.

### Study Strengths and Limitations

A key strength of this study is the collaborative-participatory design and the use of the validated ASPIRE process to develop a robust and co-developed, focused training resource, which supports the ability to provide training that is “fit for purpose”. This approach has been used in a range of contexts related to health education and training (e.g., [48,64]). Consequently, APUEC enhances intrinsic motivation to engage with the materials through the relevancy of the information to clinical practice and interactive activities that reiterate key learning and maximise engagement. APUEC is highly accessible training, which can be re-accessed and repeated, giving opportunities for end-users to review and consolidate their learning at any time. While SBIRT training exists in a variety of delivery formats (e.g., [65,66,67]), the time-poor, highly pressurised environment of UEC means that healthcare professionals may experiences challenges with accessing training around shift work and clinical demands. Workforce training for UEC workers can, therefore, be inconsistent and fragmented. The provision of brief, accessible, digital training resources, such as APUEC, can offer significant flexibility for individual completion at a time and place to suit the end-user. This has been demonstrated previously since digital resources are commonly used for the delivery of education to the emergency care workforce, in diverse areas (e.g., nursing triage [68], nurse airway assistants [69], oxygen therapy [70], detection of child abuse [71], and assessment of patients at risk of violence [72]).

A strength of the evaluation is that we assessed change at three levels of the New World Kirkpatrick Model, whereas many applications of this framework in health education only measure levels 1 and 2 (e.g., [73,74,75,76,77]). It was a pragmatic decision not to measure objective knowledge change due to time constraints for the delivery and evaluation of APUEC as one element of a training day for health champions. Therefore, we do not know whether objective knowledge levels changed due to using the package; however, as in [48], assessing factual knowledge change was not an objective of our study. Our primary aim, therefore, was to establish whether perceived knowledge, confidence and skills relating to SBIRT were greater on completion of the training than immediately before exposure to the package. Our qualitative interview data allowed us to conduct “ipsative assessment” via discussion about the training with participants to ascertain whether and how learning could be implemented in practice. Confidence in one’s skills is related to perceived knowledge and not just factual knowledge [78]. Nonetheless, the authors have since developed a pre-post knowledge questionnaire that will be used in future evaluations of the APUEC training.

Although we collected data on participants’ occupation, we did not collect data on their level of education and training or prior experience in health promotion practice, albeit all were in roles that involved health promotion. It should be recognised that individuals that attended the training and took part in the interviews were health improvement champions at their employing hospital trust and, therefore, they were likely to have had pre-existing positive attitudes towards health promotion (broadly) and engagement with alcohol misuse prevention in UEC settings (specifically). The study did not account for any potential bias in their pre-existing attitudes. It could potentially be more challenging to engage staff in APUEC training and SBIRT practice who have less positive attitudes towards health promotion at the outset. Nonetheless, APUEC training begins with a strong rationale for the focus on promoting population health (and specifically alcohol misuse prevention), and this aims to foster positive attitudes towards health promotion and SBIRT in all training recipients. Finally, evaluation data were collected immediately after participants had accessed APUEC. As such, we were unable to assess Kirkpatrick Level 4 which was beyond the scope of this study. Assessment of Level 4 might focus on the direct performance outcomes of the APUEC training, for example, any changes in clinician’s behaviour (i.e., SBIRT practices) and any resulting outcomes for patients (e.g., health behaviours, individual health and wellbeing, and UEC attendances). Few studies of digital learning resources have examined the effectiveness of e-learning on clinician behaviour and patient outcomes [76]. The longer-term impact of APUEC training on clinicians’ behaviour, and any associated health, clinical and service outcomes, is not yet known but is an area for future research.

## 5. Conclusions

APUEC makes a step-change in the provision of workforce training relating to SBIRT in UEC settings. This accessible digital training increases healthcare professionals’ perceived knowledge, confidence and skills related to alcohol prevention in UEC settings. Healthcare professionals view APUEC training as a valuable contributor to facilitating health promotion practice around alcohol prevention in UEC settings. With the focus of APUEC training on the rationale for, and delivery of, SBIRT for alcohol prevention, APUEC could make a significant contribution to workforce training in health improvement. Ultimately, this could facilitate the integration of SBIRT into routine care, which may contribute to population health improvement. Overall, we recommend that APUEC training is embedded within education and training programmes for healthcare professionals and healthcare trainees of any discipline. Further research is needed to explore mechanisms for the implementation of APUEC into workforce training programmes within healthcare organisations, end-users’ experiences of translating their learning into health promotion practices and any outcomes of for patients and healthcare organisations.

## Figures and Tables

**Figure 1 ijerph-20-07028-f001:**
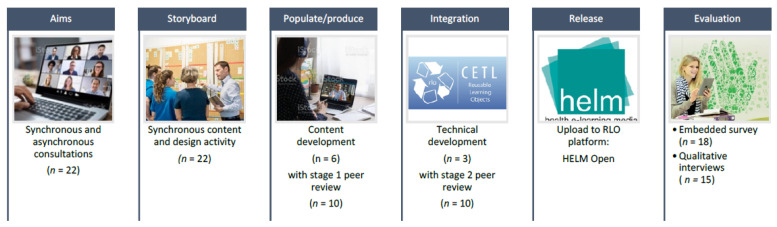
ASPIRE Methodology for digital training development.

**Figure 2 ijerph-20-07028-f002:**
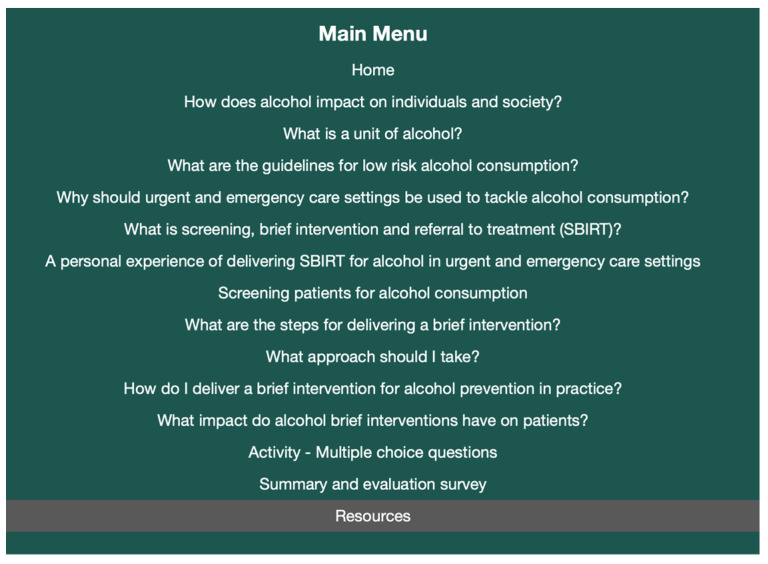
Final RLO content.

**Figure 3 ijerph-20-07028-f003:**
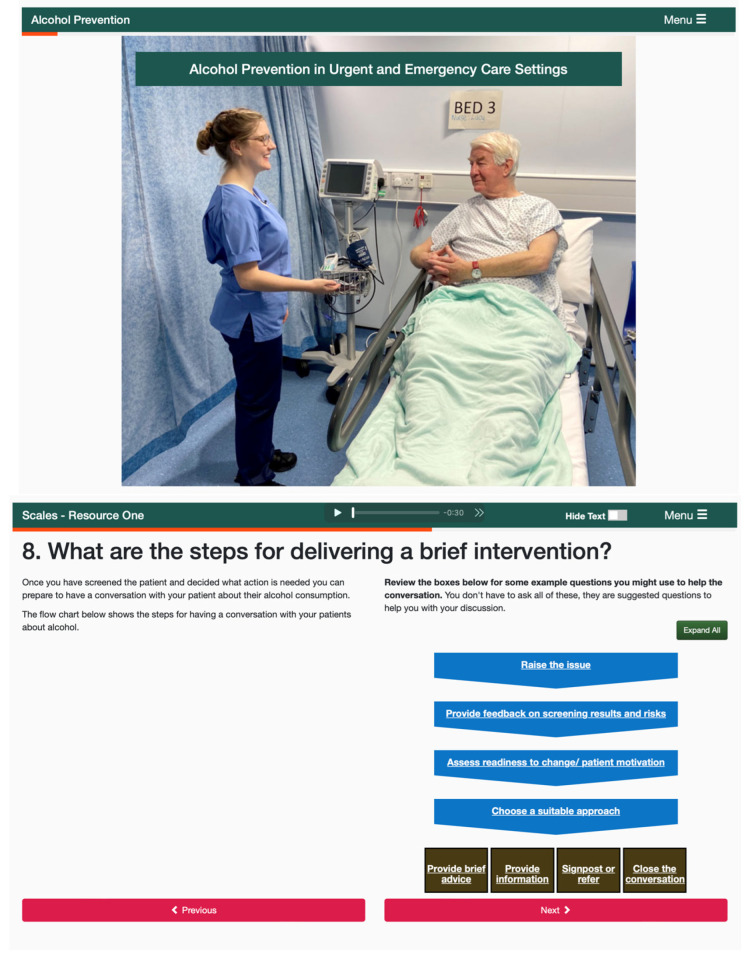
Example screenshots.

**Figure 4 ijerph-20-07028-f004:**
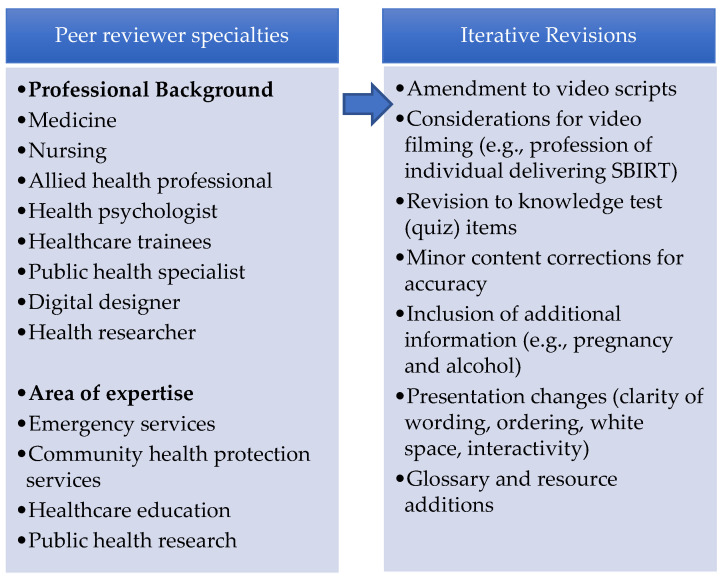
Peer reviewer details and revisions to the training resource.

**Figure 5 ijerph-20-07028-f005:**
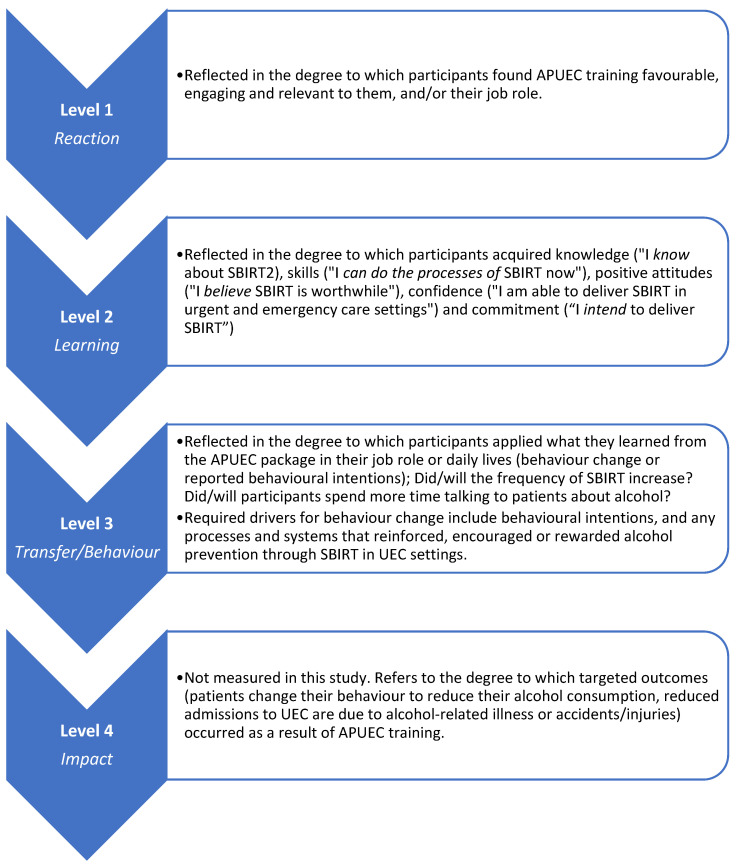
APUEC evaluation using the New World Kirkpatrick Evaluation Model.

**Table 1 ijerph-20-07028-t001:** Storyboarding questions.

Breakout Group Questions	To Consider:
Q1. What do you think is important to include in this RLO about brief interventions for alcohol prevention in urgent and emergency settings?	What are the key topics we should cover? What are the most important guidelines healthcare staff need to know about? What sort of information will be essential for urgent and emergency staff to understand to be able to deliver brief health promotion intervention around alcohol? Think about: Population (service-users); Environment; Challenges and barriers; Facilitators; Attitudes towards health promotion; Knowledge and skills; Team-working.
Q2. How do you think the information should be best presented for maximum engagement?	How best to present the content? How to make it interactive? Is there a better order for materials? What will encourage people to engage with this training?
Q3. What evidence-based resources should we signpost people to?	Extra resources aimed at staff using the RLO Helpful resources for signposting service users

**Table 2 ijerph-20-07028-t002:** Participant characteristics (gender and occupation).

ID	Gender	Occupation
101	Female	Emergency Department Assistant (EDA)
102	Male	Advanced Clinical Practitioner (ACP)
103	Female	Clinical Support Worker (CSW) Manager
104	Female	Doctor
105	Male	Doctor
106	Female	ACP/Teaching Fellow
107	Female	ACP
108	Female	Nurse
109	Female	Nurse
110	Female	Nurse
111	Female	Nurse
112	Female	Nurse
113	Female	Nurse
114	Female	Nurse
115	Male	Doctor

**Table 3 ijerph-20-07028-t003:** Measurement aligned with the New World Kirkpatrick Evaluation Framework (table adapted from [48]).

Level (1–3) ^†^	Sub-Component	Measure	Data Collection		
				Post-Survey	Interview
1	Reach	Channel for receipt of the resource User role: healthcare professional or student Geographical region		X	X
	Use	Helpfulness for learning Main reason for accessing Ease /problems with use (technical, level of difficulty, context, cultural)		X	X
	Satisfaction	Overall view and rating of the resource Elements most liked Elements least liked Recommendation to others		X	X
	Engagement	View towards interactive elements (menu, narration adjustments, video clips, information boxes, click boxes, quiz, extra resources)			X
	Relevance	Relevance to self or others Opportunity to use the resource			X
2	Knowledge	Evidence of new learning			X
	Skill	Feeling equipped with useful knowledge			X
	Attitude	Views towards APUEC training/change in views			X
	Confidence	Changes in confidence to communicate (patients or clients)			X
	Commitment	Estimated future use and resource sharing			X
3	Behaviour changes	User application of knowledge Reported behavioural changes			X X
	Required drivers	Target audiences Mechanisms for dissemination			X X

^†^ Level descriptors—Level 1: reaction; Level 2: learning; Level 3: transfer/behaviour.

**Table 4 ijerph-20-07028-t004:** Post-exposure perceptions of attitudes, knowledge, skills and confidence to engage in SBIRT.

Survey Items	N (%)
I believe patients should be screened for their alcohol consumption in UEC settings	17 (94.5)
I believe that UEC settings are suitable places to deliver brief interventions for alcohol prevention	18 (100)
I believe that brief advice from a healthcare professional can help patients to reduce their drinking and/or seek help with their drinking	16 (88.9)
I believe some patients should be referred for treatment for their alcohol consumption in urgent and emergency care settings	18 (100)
I have the knowledge to screen my patients for alcohol consumption	15 (83.3)
I know what tools to use to screen my patients for alcohol consumption	14 (77.8)
I feel confident I can screen my patients for alcohol consumption	15 (83.3)
I have the knowledge to give brief advice to my patients about reducing their alcohol consumption	14 (77.7)
I feel confident that I can give brief advice to my patients about reducing their alcohol consumption	15 (83.4)
I have the skills to give brief advice about alcohol with my patients	14 (77.8)
I intend to increase the number of patients I screen for alcohol consumption	15 (82.9)
I intend to increase the number of patients I give brief advice to about their alcohol consumption	16 (88.8)

**Table 5 ijerph-20-07028-t005:** Mixed-methods analysis aligned with the New World Kirkpatrick Evaluation Framework.

Level (1–3) ^†^	Sub-Component	Measure	N (%)
(1) Reaction	Reach	Channel for receipt of the resource A course learning resource Recommended by peer/colleague Type of User Healthcare professional *“I think everybody, all healthcare professionals, regardless of their hierarchy or their background, would benefit” [ID104, Female, Doctor]*. “*I feel like most health professionals should know about it so they can pass it on to patients, their relatives, staff”. [ID103, Female, CSW manager]*.	11 (61.1) 8 (44.4) 18 (100)
	Use	Helpful or very helpful rating: Problems with use (% yes) No problems Technical issues Level of difficulty Language difficulty Contextual or cultural differences Other issues (e.g., personal device issue, lack of time to complete) *“this training was very structured and it’s standardised” [ID104, Female, Doctor]*. *“succinct enough that they kept my attention…. the fact they had transcripts there, that was great” [ID102, Male, ACP]*. *“it was really good with the voiceovers as well… I sometimes struggle with my reading, so actually having it to listen to was really helpful” [ID113, Female, Nurse]*.	18 (100) 16 (88.9) 2 (11.1) 0 (0.0) 0 (0.0) 0 (0.0) 0 (0.0)
	Satisfaction	Would recommend to others: *“I think it’s invaluable” [ID105, Male, Doctor]*. *“I really enjoyed doing it” [ID112, Female, Nurse]*. *“it’s really been educative, and you know, it stimulates the way one learns quickly … it’s something that everyone would be happy to do any time” [ID115, Male, Doctor]*.	18 (100)
	Engagement	View towards interactive elements: *“it’s been quite informative and quite interactive” [ID108, Female, Nurse]*. *“the use of video, the use of quizzes” [ID105, Male, Doctor]*. *“I think you remember it more when you’re actively doing something” [ID112, Female, Nurse]*.	-
	Relevance	Relevance to self or others: *“very relevant, I think in A&E … we get so many alcohol related injuries in the whole population…from the students right through to the elderly” [ID108, Female, Nurse]*. *“it is something we deal with every day, like multiple of our patients in our teams will be alcohol related or drug related” [ID110, Female, Nurse]*.	-
(2) Learning	Knowledge	Learned something new: *“I like the kind of the tools that were involved. Yeah, it gave me some food for thought” [ID102, Male, ACP]*. *“the reference to the AUDIT-C umm tool for screening for alcohol. Pretty simple questions, really nice stratification of risk” [ID105, Male, Doctor]*. *“I know how to easily … keep on track, engage with them, keep on track with the conversation because it’s all straight in my head” [ID115, Male, Doctor]*.	-
	Skill	Feeling equipped with useful knowledge: *“when I’m talking to patients or colleagues … about their alcohol, about their relationship and its potential impact, I think it will help … give me a bit more structure, which I’m not doing now … how I approach the subject and allow them to talk so we can move through it together” [ID102, Male, ACP]*. *“I’m learning to even incorporate all of those social determinants of health just to find out and yes, it does give us a lot of information to me, as a doctor to decide and help personalise care for this patient based on their individual circumstances” [ID104, Female, Doctor]*. *“it was a good resource to learn about how to initiate that conversation with people who aren’t necessarily being admitted to ED for alcohol use. So I thought that that aspect of it was quite handy cause it is a bit of an awkward conversation to have, isn’t it?” [ID110, Female, Nurse]*. *“it’s given me more of an insight into what exactly to ask to cut out all the ‘gobbledygook’ and just get to the point. But at the same time have that patient relationship but know exactly what the important questions are to ask as opposed to going through a whole quiz about drinking” [ID109, Female, Urgent care practitioner]*.	-
	Attitude	Views towards alcohol prevention and/or SBIRT: *“we have to start talking about health improvement” [ID110, Female, Nurse]*. *“I hope it empowers people to that, you know, actually, we’re all responsible for having these conversations, and we all can have an impact on a patient’s health and well-being. So we should be having these conversations” [ID106, Female, ACP]*. *“I think it should be less of a taboo and I think the more we have these conversations with patients, the easier it comes for us just to make it into our, like our normal” [ID107, Female, ACP]*. *“I think A&E is a great place to kind of capture people and make…meaningful kind of adjustments or impacts” [ID102, Male, ACP]*. *“if we’ve got people with better health kind of knowledge it could lead to better outcomes. So ultimately it leads to a reduced stress on the system. Potentially” [ID102, Male, ACP]*.	-
	Confidence	Increased confidence to deliver SBIRT: *“I think once you’ve had that extra training, you’ve got the knowledge base and you know where to signpost people” [ID108, Female, Nurse]*. *“it just helps them [staff] become better communicators with our patients, you know, like the videos making sure that we’re not, we’re not kind of coming across as judgmental” [ ID106, Female, ACP]*. *“…had I received that, that teaching, that training, looked at that resource, six, seven, eight years ago when I was a more junior member of staff, absolutely it would have given me the confidence” [ID106, Female, ACP]*. *“it has reinforced me in, in having this confidence that whatever I am doing and the approach that I have had so far” [ID104, Female, Doctor]*. *“it’s giving me more confidence and understanding” [ID107, Female, ACP]*. *“I feel, I feel a lot more comfortable talking about it” [ID111, Female, Nurse]*.	-
	Commitment	Estimated future use and resource sharing: *“that’s really good. I’ll implement that, that’s a really simple thing I can do” [ID112, Female, Nurse]*. *“I think I would want to be able to share it to perhaps other people. If they were like learning how to give out advice, absolutely I think it would probably benefit a lot of people” [ID106, Female, ACP]*.	-
(3) Transfer/ Behaviour	Behavioural intention and/or behavioural changes	User application of knowledge and reported behavioural intentions and/or changes: *“I’ll be referring, referring them to alcohol specialists or the teams that we have on site” [ID101, Female, EDA]*.	-
	Required drivers	Target audiences and mechanisms for dissemination (i.e., who should use SBIRT, approaches for transfer of learning into practice, and when should it happen). *“It should be everyone… who has a contact to the patient and depending on who, who is able to see the patient first” [ID115, Male, Doctor]*. *“just everybody because I think everybody has got the opportunity to, to give that advice even if it’s just 5 min” [ID106, Female, ACP]*. *“it’s the approachability of that person. So if, like the doctor says, well I’ve tried to have this conversation with this patient, would you mind just going in and seeing if you can get them to open up a little bit more, if we support each other within the wider team” [ID110, Female, Nurse]*. *‘sometimes the quiet 10 min chat you get is when you’ve taken a patient round to X-ray. So that could be a nurse, an EDA, CSW” [ID113, Female, Nurse]*. *“Like we work together as a unit, I feel like that would be quite a good way to get rid of those kind of barriers” [ID110, Female, Nurse]*.	-

^†^ Level descriptors—Level 1: reaction; Level 2: learning; Level 3: transfer/behaviour ACP: advanced clinical practitioner; EDA: emergency department assistant; CSW: clinical support worker; A&E: accident and emergency department.

## Data Availability

The data and materials that support the findings of this research are made available from the corresponding author upon reasonable request.

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
