# Peer review of "Alcohol Prevention in Urgent and Emergency Care (APUEC): Development and Evaluation of Workforce Digital Training on Screening, Brief Intervention, and Referral for Treatment"

_ijerph, 2023, doi:10.3390/ijerph20227028_

Round 1
Reviewer 1 Report
Comments and Suggestions for Authors
The content of the article is highly relevant to the scientific field and health promotion in relation to alcohol use intervention. The introduction includes the presentation of current and relevant references to the field. I only suggest including the relevance of the approach, considering the great demand for emergencies related to alcohol abuse, for example: the amount of demand related to this.
Clear and objective method and results. Justify the selection of the sample of professionals who evaluated the strategy, as it shows bias in the analysis.
Reviewer 2 Report
Comments and Suggestions for Authors
This manuscript describes development and testing of digital education system on preventing alcohol abuse for health professionals. The system was co-developed with health professionals.
Testing was done by survey and qualitative interviews.
The strength of the study is sound framework (ASPIRE process or Kirkpatrick model).
The potential weakness of the study is that interview with on-board members may not reflect responses from real world users.
1) The potential that interview with on-board members may not reflect responses from real world users can be discussed.
2) Table 5:
Rows of measure and N are not aligned. Alignment should be adjusted.
3) Table 5:
Why n is not presented in lower rows?
Reviewer 3 Report
Comments and Suggestions for Authors
The abstract effectively communicates the importance and effectiveness of the APUEC digital training package in addressing the critical issue of excessive alcohol consumption in UEC settings. This approach has the potential to empower healthcare professionals with the necessary tools to make a meaningful impact on patient well-being and population health.
Introduction
- The introduction is quite long and contains a lot of information. Consider breaking it down into subsections to make it more reader-friendly. For instance, you could have separate sections for the background on alcohol consumption, the role of UEC settings, barriers to SBIRT implementation, and the need for training.
- Some claims in the introduction are not supported by citations. You may consider adding more citations.
- The introduction repeats some points about healthcare professionals' barriers to SBIRT delivery. Consider removing some of the redundancy to maintain conciseness.
- Provide specific examples or case studies that illustrate the issues raised. For instance, you mention that some recipients suggest that the approach, timing, or delivery of SBIRT could be improved in the emergency context. It would be helpful to cite specific examples or studies that support this claim.
- Elaborate further on why digital training is chosen as the medium for addressing the identified barriers. What advantages does a digital approach offer over traditional training methods? Provide evidence or references to support this choice.
- Acknowledge potential challenges in implementing APUEC, such as technology access and acceptance among UEC workers, and discuss how these challenges might be addressed.
- Consider adding a brief concluding sentence to this introduction to clearly transition to the subsequent sections of the paper.
Methods
- The methods section is well-structured, and the steps in the development and evaluation process are clearly delineated. The use of headings and subheadings aids in easy navigation.
- While the meanings of acronyms like RLO, ASPIRE, and TIDieR are explained, it would be helpful to include a list of abbreviations at the beginning for quick reference.
- The section describing RLOs and ASPIRE methodology is informative and helps the reader understand the foundation of the digital training package. It might be beneficial to provide references or examples of previous studies that have successfully employed RLOs and ASPIRE methodology.
- The timeline of development and survey evaluation is clearly stated. However, consider providing more context on the choice of timeframes. Why was this specific period chosen for each stage? Are there any seasonal or contextual factors that influenced these timelines?
- The description of the peer review process is insightful, but it could be enhanced by detailing the specific feedback received and how it influenced revisions. Were there any common themes in the feedback that required multiple revisions?
- While you mention the 15 interview participants, consider providing more information on the selection criteria and how participants were recruited. This would help in understanding the representativeness of the sample.
- The New World Kirkpatrick Evaluation Model is explained well, and the measurement indicators for levels 1-3 are detailed. However, it might be helpful to explicitly state that Level 4 (Impact) is not measured in this study. Additionally, consider briefly discussing the rationale for focusing on Levels 1-3 and the limitations associated with not assessing Level 4.
- Include more details on the quantitative survey items and the specific questions asked during the qualitative interviews. This will provide transparency and clarity on how the data were collected.
-Relate the methods back to the research questions outlined in the introduction. Explain how each step in the methods contributes to addressing these research questions.
Results
The section provides a comprehensive overview of participant reactions, learning outcomes, and behavioral intentions, offering insights into various aspects of the study.
- While participant quotes are valuable, consider providing a more structured approach to summarizing qualitative data, such as themes or categories that emerged from the interviews. This can help readers quickly grasp the main qualitative findings.
- Consider additional analysis based on the qualitative data, such as a thematic analysis of participant quotes. This could offer more in-depth insights into their perceptions and experiences.
Discussion
The Discussion section effectively highlights the significance of the study by emphasizing its contribution as the first evidence-based digital training package for UEC workers. It's clear that the training, 'Alcohol Prevention in Urgent and Emergency Care' (APUEC), addresses a critical gap in healthcare training.
- The discussion about the engagement and accessibility features of APUEC is well-presented, and the inclusion of UK Web Content Accessibility Guidelines (WCAG) standards is valuable. However, while the section addresses the benefits of these features, it could be more explicit in explaining how these aspects improve training outcomes.
- The discussion acknowledges the potential for further digital training for the UEC workforce, which is a valuable point. However, it could be more explicit in suggesting specific research questions or areas for exploration in future studies.
- While the section mentions the barriers to SBIRT implementation in UEC settings, consider providing more detailed insights into each of these barriers. For example, delve deeper into the structural challenges, time constraints, and privacy issues that healthcare professionals face in these environments. This can help readers better understand the context.
- The discussion mentions that the study did not assess objective knowledge levels. Consider explaining the rationale behind this choice. Why wasn't factual knowledge change an objective, and what are the implications of this decision for the study's findings?
- When discussing the positive attitudes of the health improvement champions, address how the study accounted for the potential bias in their pre-existing attitudes. Are there any measures in place to mitigate the influence of participants' prior positive attitudes?
- Discuss the study's limitations regarding the longer-term impact of APUEC training on clinician behavior and patient outcomes. This can help readers understand the scope of the study and its potential for future research.
- Even though Level 4 evaluation was beyond the scope of the study, it might be valuable to mention what a Level 4 evaluation would entail and its importance for assessing the real-world impact of the training. This can provide context for future research.
Comments on the Quality of English Language
The abstract effectively communicates the importance and effectiveness of the APUEC digital training package in addressing the critical issue of excessive alcohol consumption in UEC settings. This approach has the potential to empower healthcare professionals with the necessary tools to make a meaningful impact on patient well-being and population health.
Introduction
- The introduction is quite long and contains a lot of information. Consider breaking it down into subsections to make it more reader-friendly. For instance, you could have separate sections for the background on alcohol consumption, the role of UEC settings, barriers to SBIRT implementation, and the need for training.
- Some claims in the introduction are not supported by citations. You may consider adding more citations.
- The introduction repeats some points about healthcare professionals' barriers to SBIRT delivery. Consider removing some of the redundancy to maintain conciseness.
- Provide specific examples or case studies that illustrate the issues raised. For instance, you mention that some recipients suggest that the approach, timing, or delivery of SBIRT could be improved in the emergency context. It would be helpful to cite specific examples or studies that support this claim.
- Elaborate further on why digital training is chosen as the medium for addressing the identified barriers. What advantages does a digital approach offer over traditional training methods? Provide evidence or references to support this choice.
- Acknowledge potential challenges in implementing APUEC, such as technology access and acceptance among UEC workers, and discuss how these challenges might be addressed.
- Consider adding a brief concluding sentence to this introduction to clearly transition to the subsequent sections of the paper.
Methods
- The methods section is well-structured, and the steps in the development and evaluation process are clearly delineated. The use of headings and subheadings aids in easy navigation.
- While the meanings of acronyms like RLO, ASPIRE, and TIDieR are explained, it would be helpful to include a list of abbreviations at the beginning for quick reference.
- The section describing RLOs and ASPIRE methodology is informative and helps the reader understand the foundation of the digital training package. It might be beneficial to provide references or examples of previous studies that have successfully employed RLOs and ASPIRE methodology.
- The timeline of development and survey evaluation is clearly stated. However, consider providing more context on the choice of timeframes. Why was this specific period chosen for each stage? Are there any seasonal or contextual factors that influenced these timelines?
- The description of the peer review process is insightful, but it could be enhanced by detailing the specific feedback received and how it influenced revisions. Were there any common themes in the feedback that required multiple revisions?
- While you mention the 15 interview participants, consider providing more information on the selection criteria and how participants were recruited. This would help in understanding the representativeness of the sample.
- The New World Kirkpatrick Evaluation Model is explained well, and the measurement indicators for levels 1-3 are detailed. However, it might be helpful to explicitly state that Level 4 (Impact) is not measured in this study. Additionally, consider briefly discussing the rationale for focusing on Levels 1-3 and the limitations associated with not assessing Level 4.
- Include more details on the quantitative survey items and the specific questions asked during the qualitative interviews. This will provide transparency and clarity on how the data were collected.
-Relate the methods back to the research questions outlined in the introduction. Explain how each step in the methods contributes to addressing these research questions.
Results
The section provides a comprehensive overview of participant reactions, learning outcomes, and behavioral intentions, offering insights into various aspects of the study.
- While participant quotes are valuable, consider providing a more structured approach to summarizing qualitative data, such as themes or categories that emerged from the interviews. This can help readers quickly grasp the main qualitative findings.
- Consider additional analysis based on the qualitative data, such as a thematic analysis of participant quotes. This could offer more in-depth insights into their perceptions and experiences.
Discussion
The Discussion section effectively highlights the significance of the study by emphasizing its contribution as the first evidence-based digital training package for UEC workers. It's clear that the training, 'Alcohol Prevention in Urgent and Emergency Care' (APUEC), addresses a critical gap in healthcare training.
- The discussion about the engagement and accessibility features of APUEC is well-presented, and the inclusion of UK Web Content Accessibility Guidelines (WCAG) standards is valuable. However, while the section addresses the benefits of these features, it could be more explicit in explaining how these aspects improve training outcomes.
- The discussion acknowledges the potential for further digital training for the UEC workforce, which is a valuable point. However, it could be more explicit in suggesting specific research questions or areas for exploration in future studies.
- While the section mentions the barriers to SBIRT implementation in UEC settings, consider providing more detailed insights into each of these barriers. For example, delve deeper into the structural challenges, time constraints, and privacy issues that healthcare professionals face in these environments. This can help readers better understand the context.
- The discussion mentions that the study did not assess objective knowledge levels. Consider explaining the rationale behind this choice. Why wasn't factual knowledge change an objective, and what are the implications of this decision for the study's findings?
- When discussing the positive attitudes of the health improvement champions, address how the study accounted for the potential bias in their pre-existing attitudes. Are there any measures in place to mitigate the influence of participants' prior positive attitudes?
- Discuss the study's limitations regarding the longer-term impact of APUEC training on clinician behavior and patient outcomes. This can help readers understand the scope of the study and its potential for future research.
- Even though Level 4 evaluation was beyond the scope of the study, it might be valuable to mention what a Level 4 evaluation would entail and its importance for assessing the real-world impact of the training. This can provide context for future research.
Reviewer 4 Report
Comments and Suggestions for Authors
This manuscript provides valuable insights into the development and evaluation of a workforce digital training program for alcohol prevention in urgent and emergency care settings. However, incorporating the solutions to the following points would further strengthen the quality of this work:
1. Abstract:
Lines 18 - 39: The abstract lacks sufficient background information on the significance of excessive alcohol consumption and the current challenges in integrating SBIRT into urgent and emergency care (UEC) settings. Providing more perspectives would help readers understand the problem being addressed and the relevance of the study. While the abstract mentions the development, delivery, and evaluation of the 'Alcohol Prevention in Urgent and Emergency Care' (APUEC) digital training package, it does not provide enough detail on the specific components, structure, or duration of the training. This information would give readers a better understanding of the intervention being evaluated. The abstract briefly mentions that the APUEC digital training increases healthcare professionals' perceived knowledge, confidence, and skills related to alcohol prevention in UEC settings. However, it does not provide specific details or quantitative results to support this claim. Adding more specific findings would enhance the credibility and impact of the study. The abstract mentions the recommendation to embed the digital training in education and professional development programs for healthcare professionals and trainees but does not elaborate on potential implementation strategies or challenges. Expanding on these recommendations would provide readers with actionable insights and increase the practical relevance of the study. Further, avoid the usage of personal pronouns and consider minimizing the usage of acronyms in the abstract.
Line 26: Kindly check the term “UECS”
2. Introduction:
Lines 43 – 116: The introduction lacks a clear and organized structure, making it difficult for readers to follow the flow of information. Consider dividing the introduction into subsections to address specific aspects, such as the global burden of alcohol consumption, the need for alcohol prevention in urgent and emergency care settings, and the barriers to implementing SBIRT. While the introduction provides some statistics and highlights the negative consequences of alcohol consumption, it lacks a comprehensive overview of the current state of alcohol use, alcohol-related burden, and existing interventions. Adding more context would help readers understand the significance and relevance of the study. There is repetition of information throughout the introduction. For example, the positive attitudes of healthcare professionals towards health promotion and alcohol prevention in urgent and emergency care settings are mentioned twice within a short span. Streamlining the content and avoiding unnecessary repetition would improve the clarity and conciseness of the introduction.
Although several references are provided, there is room for including more recent (2022 and 2023) and relevant studies to support the statements made in the introduction. This would strengthen the credibility of the research and demonstrate a comprehensive understanding of the existing literature. While the introduction mentions the need for workforce training on alcohol prevention and SBIRT in urgent and emergency care settings, it does not clearly explain the specific research gap that the study aims to address. Clearly articulating the gap in the literature would help readers understand the novelty and importance of the study. The research questions listed at the end of the introduction are vague and could be more specific. Providing clearer research questions would enhance the understanding of the study's objectives and facilitate a more focused approach to data collection and analysis.
3. Materials and Methods:
Lines 117 -257: This section does not clearly mention and clarify the number of participants involved in the development and evaluation stages of the digital training package. It's important to provide this information to ensure transparency and reproducibility of the study. While this section mentions that the survey items were adapted from the 'Evaluation Toolkit for Reusable Learning Objects and deployment of e-Learning Resources,' it would be helpful to provide more details on the specific items used and how they were modified for this study. This would enable readers to understand the content and relevance of the survey. This section briefly mentions that qualitative interviews were conducted with a subset of training recipients, but does not provide sufficient details on the qualitative aspects of the interview process. It would be beneficial to include information on the interview protocol, the number of interviews conducted, and any specific themes or questions explored during the interviews. This section does not explicitly discuss the limitations of the study. It would be valuable to address potential limitations, such as any biases in participant selection or the generalizability of the findings, to provide a balanced interpretation of the results.
Line 137: Acronyms should be defined during its first time appearance in the main running text. E.g. RLO is not defined.
Page 4: Visual representation and quality of figure 1 could be improved.
Page 5 - Line 190: Table 1: The presentation of the “Breakout Group Questions” and “To consider” could be improved.
Page 6: The content and presentation of figure 2 could be improved.
Lines 223-225: The visual representation of figure 3 could be improved; the some of the contents are not clearly visible. Consider labelling the vital information in this diagram.
Line 227: The caption/title of figure 4 could be improved.
Line 249: The caption/title of table 2 could be improved with specific details.
4. Results:
Line 264 - 398: The results section does not provide information on the characteristics of the participants, such as their levels of experience. This information could be crucial in understanding the generalizability of the findings and the potential impact of the training on different healthcare professionals.
Lines 392: In table 4, would it be possible to evaluate the statistical significance of data presented.
Lines 394: In table 5, would it be possible to evaluate the statistical significance of data presented.
5. Discussions:
Lines 399 – 510: The discussion section repeats certain points multiple times, such as the accessibility features of the digital training and the need for addressing barriers to implementation. While it's essential to reiterate important information, excessive repetition may make the section less concise and engaging. The discussion primarily focuses on the strengths and positive aspects of the study, particularly the benefits of the developed digital training. It would be beneficial to include a more balanced analysis by discussing potential limitations and areas for improvement. The study emphasizes the positive feedback received from participants who were health improvement champions in their hospital trusts. However, this may limit the generalizability of the findings to a broader range of healthcare professionals who may have different attitudes towards health promotion and alcohol misuse prevention. While the study evaluates the immediate impact of the digital training on participants, it acknowledges the inability to assess the long-term impact (Kirkpatrick Level 4) due to the scope of the study. It would be valuable to discuss potential plans or recommendations for future research to assess the sustained effects of the training.
6. Conclusions:
Lines 511 – 526: The conclusions could benefit from providing more specific details about the impact and benefits of the APUEC training. Instead of using general statements like "increases healthcare professionals' perceived knowledge, confidence, and skills," it would be helpful to quantify the extent of improvement or provide specific examples of how the training positively impacted participants. While APUEC is highlighted as a valuable training resource, the conclusions section does not adequately address potential alternatives or discuss the limitations of relying solely on this training. It would be beneficial to acknowledge that APUEC may not fully address all training needs or preferences, and that a variety of training approaches should be considered. The conclusions focus on recommending the embedding of APUEC training within education and training programs. However, there is no discussion about the scalability of the training or the potential challenges that may arise when implementing it on a larger scale. It would be important to address the feasibility and practicality of widespread integration. While the conclusion mentions the need for further research, it does not provide specific research questions or areas that should be explored. It would be helpful to include more concrete recommendations for future studies, such as investigating the long-term impact of APUEC on patient outcomes or conducting comparative studies with other training approaches.
7. References:
Consider improving the presentation of the references and citations throughout the manuscript.
Comments on the Quality of English Language
Moderate editing of English language required
Round 2
Reviewer 3 Report
Comments and Suggestions for Authors
All comments were addressed.
Comments on the Quality of English LanguageFew minor edits are needed.
Reviewer 4 Report
Comments and Suggestions for Authors
The authors have provided solutions to most of the comments.
Comments on the Quality of English LanguageMinor English language editing is required.